# Ablation of non-coding RNAs affects bovine leukemia virus B lymphocyte proliferation and abrogates oncogenesis

**Roghaiyeh Safari**[1,2], **Jean-Rock Jacques**[1,2], **Yves Brostaux**[3], **Luc Willems**[1,2]*

**1** Molecular and Cellular Epigenetics (GIGA), University of Liege, Sart-Tilman Liège, Belgium, **2** Molecular and Cellular Biology (TERRA), Gembloux Agro-Bio Tech, University of Liege, Gembloux, Belgium, **3** Modelisation and development, Gembloux Agro-Bio Tech, University of Liege, Gembloux, Belgium

* Luc.willems@uliege.be

**Data Availability Statement:** All relevant data are within the manuscript and its Supporting Information files.

## Abstract

Viruses have developed different strategies to escape from immune response. Among these, viral non-coding RNAs are invisible to the immune system and may affect the fate of the host cell. Bovine leukemia virus (BLV) encodes both short (miRNAs) and long (antisense AS1 and AS2) non-coding RNAs. To elucidate the mechanisms associated with BLV non-coding RNAs, we performed phenotypic and transcriptomic analyzes in a reverse genetics system. RNA sequencing of B-lymphocytes revealed that cell proliferation is the most significant mechanism associated with ablation of the viral non-coding RNAs. To assess the biological relevance of this observation, we determined the cell kinetic parameters in vivo using intravenous injection of BrdU and CFSE. Fitting the data to a mathematical model provided the rates of cell proliferation and death. Our data show that deletion of miRNAs correlates with reduced proliferation of the infected cell and lack of pathogenesis.

## Author summary

BLV is a retrovirus that integrates into the genomic DNA of B-lymphocytes from a series of ruminant species (cattle, sheep, zebu, water buffalo and yack). Expression of viral proteins is almost undetectable in infected animals. In contrast, the BLV genome contains a cluster of 10 microRNAs that are abundantly transcribed in BLV-infected cells in vivo. In this report, we show that these microRNAs primarily regulate host cell proliferation. Ablation of the viral microRNAs affects BLV replication and suppresses leukemia development.

## Introduction

Bovine leukemia virus (BLV) is a retrovirus that naturally infects cattle, water buffalo, yak and zebu [1]. Except in Europe, BLV is a globally widespread pathogen causing significant economic losses [2, 3]. Although BLV infection is typically asymptomatic, about one third of the infected animals develop persistent lymphocytosis (PL), a stable polyclonal proliferation of

**Funding:** This work was supported by the Belgian National Fund for Scientific Research (FNRS T006014 and T026120), the Belgian Foundation against Cancer, the Télévie and the "Fondation Léon Fredericq". RS is a fellow of the Belgian Foundation against Cancer and the Télévie. LW is research director of the FNRS. The funders had no role in study design, data collection and analysis, decision to publish, or preparation of the manuscript.

**Competing interests:** The authors have declared that no competing interests exist.

non-transformed B-lymphocytes in the peripheral blood [4]. After long latency periods (7–10 years), approximately 5–10% of infected animals will die from a neoplastic B cell disease called enzootic bovine leukemia/lymphoma [5]. BLV can also be transmitted to sheep, in which the pathogenesis occurs after shorter latency periods (approximately 2–3 years) and with higher incidence (up to 100%) [6, 7].

Experimental evidence indicates that viral replication is tightly controlled by an effective immune response [8, 9]. First, ablation of lymphoid organs (i.e. spleen) accelerates pathogenesis [10]. Second, transient stimulation of viral expression ex vivo reduces life expectancy of B cells in vivo. Third, cyclosporine treatment indicates that an efficient immune response is required to control virus-expressing cells [11]. Fourth, a massive depletion of infected B cells occurs concomitantly with initiation of an immune response [12]. And finally, activation of viral expression with an HDAC inhibitor reduces the proviral loads and clears leukemia [13, 14]. Consistently, expression of viral proteins is repressed to almost undetectable levels by a still unknown mechanism [15–17]. In contrast, the BLV genome encodes viral miRNAs via internal RNA polymerase III (RNAPIII) promoters and antisense transcripts (AS1S/L and AS2) from the 3'LTR [18–20]. Located at the 5' end of the X region just downstream of the envelope gene, a cluster of five miRNAs hairpins is transcribed from a canonical type 2 RNAPIII promoters also driving tRNA synthesis [18, 19, 21]. Consistently with a high level of expression, the genomic DNA encompassing the miRNA cluster contains hypomethylated CpG. This organization in cluster leads to Drosha independent miRNA processing [22]. The RNAPIII promoter elements include characteristic A-B box sequences as transcription start site, transcription factor binding sites and a poly-T terminator [18, 22]. Thus, each BLV pre-miRNA is directly transcribed by RNAPIII giving rise to abundant expression of miRNAs in both leukemic and nonmalignant clones [23]. High levels of viral miRNAs are also found in the plasma of BLV-infected cows [24], suggesting a mechanism of paracrine signaling. Although polymorphisms were identified, the sequence of the miRNA cluster is well conserved among isolates, particularly in the seed region [20, 22, 25, 26]. Target genes whose transcripts are affected by the miRNAs have been identified. For example, miR-B4 is an analog to the host oncogenic miRNA miR-29 [18]. RNA sequencing of peripheral blood mononuclear cells (PBMCs) revealed the complexity of miRNA targets in the bovine species. BLV miRNAs modulate the expression of genes involved in oncogenesis, cell signaling, apoptosis and immune response [24]. In particular, miR-B4 targets FOS, GZMA and PPT1 RNAs were validated by luciferase reporter assays. The serine protease GZMA expressed mostly by natural killer (NK) cells and cytotoxic T-lymphocytes (CTL) but also by B cells under inflammatory conditions induces caspase-independent apoptosis. FOS mediates the primary response to B-cell receptor signaling upon dimerization with c-JUN in the AP1 complex. PPT1 removes thioester-linked fatty acyl groups from cysteine residues and modulates TNFalpha signaling. Further characterization of the multiple interactions between the BLV miRNAs with the host transcriptome will help to understand the complexity of the mechanisms involved in the bovine species.

The BLV provirus also constitutively expresses alternatively spliced transcripts (AS1 and AS2) from the antisense strand [20]. The AS1 RNA can be alternately polyadenylated, generating two transcripts AS1-S and AS1-L that are retained in the nucleus suggesting a lncRNA-like role. The AS1-L transcript overlaps the microRNA cluster and is cleaved by the RNA-induced silencing complex (RISC) [20].

Although the function of these non-coding RNAs is still unknown, deletion of the miRNA cluster from an infectious BLV molecular clone reduces viral replication in the bovine species [24]. Short-term follow-up in the ovine model indicates that pathogenesis may also be affected in absence of viral miRNAs. To understand the mechanisms involved, we analyzed the transcriptome of sorted B cells and quantified the cell turnover in vivo using a reverse genetics model.

## Results

### Oncogenesis is abrogated in the absence of BLV miRNAs

Using a reverse genetics system, we have previously shown that ablation of BLV miRNAs is dispensable for infectivity but correlates with a reduction of viral replication [24]. Indeed, the short-term proviral loads were significantly lower in animals inoculated with an isogenic provirus devoid of miRNAs (pBLV-ΔmiRNA) compared to wild-type controls (pBLV-WT). Long-term follow-up validated this conclusion in the sheep animal model (p = 0.003, according to t-test, Fig 1A). Consistently, the percentages of B cells among PBMCs were significantly lower in the absence of viral miRNAs (p = 0.002, according to Mann-Whitney U test, S1A Fig). Similarly, the ability of PBMCs to spontaneously express BLV virus in culture was reduced in sheep infected with pBLV-ΔmiRNA compared to the wild-type (p = 0.01, according to Mann-Whitney U test, S1B Fig). Although there was a trend for reduced B cell counts in the absence of miRNAs, the difference was not statistically significant (p = 0.21, S1C and S1D Fig). Since proviral loads are among the best prediction markers of pathogenesis, onset of leukemia/lymphoma consistently occurred earlier in pBLV-WT injected animals (median survival of 3.23 years, p = 0.01 according to the Log-rank Mantel-Cox test, Fig 1B). In contrast, all sheep infected with pBLV-ΔmiRNA virus remained healthy up to 7 years post-inoculation.

We conclude that onset of leukemia/lymphoma in the highly susceptible sheep model requires integrity of the miRNAs.

### Transcriptomic changes occur mostly in B cells

Soon after infection, BLV mainly replicates via the production of viral particles and infection of new cells. Then, the population of infected cells undergoes a massive depletion due to a very efficient immune clearance [12]. Thereafter, viral replication almost exclusively occurs via clonal expansion of surviving cells. It is thus predicted that cell proliferation is an important parameter of viral replication. To uncover the mechanisms associated with BLV miRNAs, we performed a transcriptomic analysis in B and non-B cells isolated from pBLV-ΔmiRNA and pBLV-WT infected animals. Details of the RNA sequencing procedures are provided in the materials and methods section. Principal component analysis (PCA) of the transcriptomics data was performed by regularized-logarithm transformation (rlog) [27] (Fig 2A). PCA revealed a clear segregation between B and non-B cell populations (Fig 2A). The PCA data from non-B cells infected by pBLV-ΔmiRNA and pBLV-WT mostly overlapped, indicating similar expression profiles independently of the type of virus and proviral load. In contrast, PC1 in B cells infected by pBLV-ΔmiRNA and pBLV-WT viruses indicated a difference in gene expression profiles.

Differential gene expression was then analyzed with the Dseq2 software. To anticipate how the log fold changes, vary with respect to the average expression levels of genes, we generated plots using shrinkage of effect size (log fold change estimates). These plots confirmed that the differentially expressed genes within B cells had a greater effect-size than those within the non-B cells (Fig 2B). Considering all data, 2158 and 67 genes were significantly differentially expressed in B and non-B cells, respectively (Fig 2C). Although most changes in gene expression occurred in B cells, there were 11 common genes between two groups (S1 Table). Comparison of B cells infected by pBLV-WT and pBLV-ΔmiRNA viruses identified 797 and 1361 significantly overexpressed genes, respectively (Fig 2D).

Overall, these analyses demonstrate that transcriptomic changes occur mostly in B cells from wild-type and miRNA-deleted viruses.

A

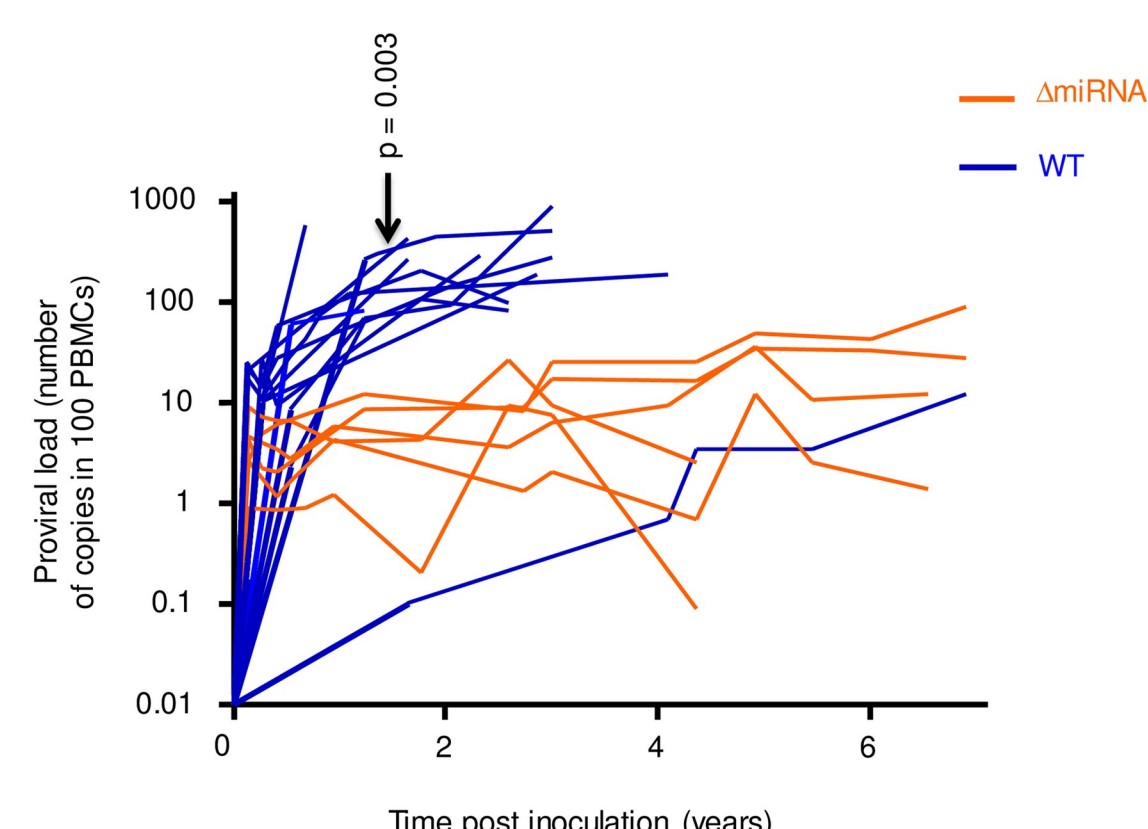

B

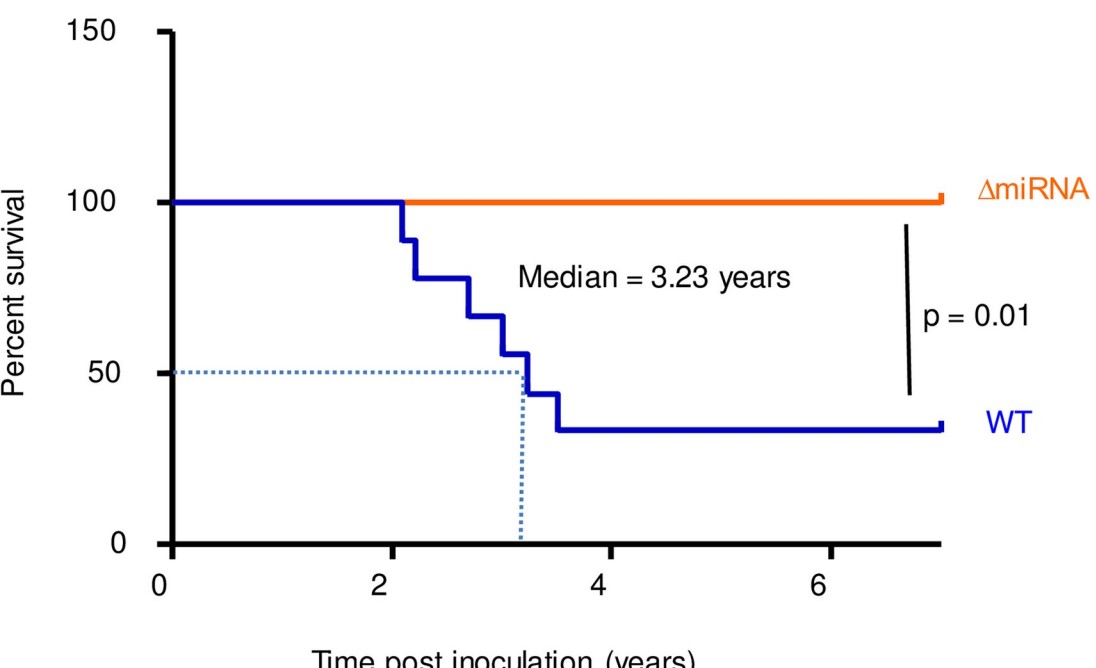

**Fig 1. Viral replication and pathogenesis in sheep infected with wild-type BLV and miRNA deletant.** (A) The proviral loads (number of copies in 100 PBMCs) were quantified in sheep infected with wild-type virus (pBLV-WT; blue lines) or miRNA-deletant (pBLV-ΔmiRNA; orange lines). p = 0.003, according to t-test. (B) The Kaplan-Meier survival curves of sheep infected with pBLV-ΔmiRNA (n = 6) and pBLV-WT (n = 12). Median survival in pBLV-WT injected animals was 3.23 years. p = 0.01, according to the Log-rank Mantel-Cox test.

## Cell division pathways are enriched in B cells from wild-type BLV infected sheep

To determine the biological functions associated with transcriptomics data, we performed an enrichment analysis of gene ontology (GO) terms using MSigDB and GSEA software [28, 29]. Analysis of the gene sets involved in biological processes (n = 4436), cellular components (n = 580) and molecular functions (n = 901) [30] yielded the enrichment map shown in Fig 3A. Based on a false discovery rates FDR<0.001, biological processes associated with cell division were significantly enriched in pBLV-WT-infected B cells (e.g. mitotic division, chromosome segregation and kinetochore in Fig 3A). This conclusion was confirmed by a statistical analysis based on a family wise-error rate FWER<0.001 (Fig 3B and enrichment plots in S2 Fig). To identify the genes driving the enrichment score in the GSEA method, we performed leading edge (LE) analysis on enriched gene sets with family wise-error rate <0.001. Chord diagrams were then generated to display the association between leading genes and enriched GO clusters [31]. Genes involved in cell mitosis (e.g. chromatid segregation, chromosome centromeric region, histone exchange, and kinetochore) were enriched in pBLV-WT-infected B cells (S3 Fig and S2 Table). Besides cell division, pathways affected by the presence of BLV miRNAs pertained to DNA repair (S3 Table). The pathways associated with the absence of miRNAs were inflammation response, immunity and cell signaling (S4 Table).

Overall, these analyses show that cell division pathways are enriched in B cells from wild-type BLV infected sheep.

## The cell turnover of peripheral blood B cells is reduced in sheep infected with pBLV-ΔmiRNA

Transcriptomic analyses thus indicated that cell division is the main mechanism that segregates B cells from wild-type and miRNA-deleted infected sheep. To unravel the biological significance of this conclusion in vivo, we analyzed the cell turnover of peripheral blood B cells. For this purpose, carboxyfluorescein succinimidyl ester (CFSE) was injected intravenously in pBLV-WT and pBLV-ΔmiRNA infected animals. Since CFSE is very unstable and only labels proteins during a short period of time, this experimental protocol provides kinetic parameters of the B cell population circulating in the bloodstream [32, 33]. Upon snapshot CFSE labeling, the fluorescent dye is progressively lost mainly due to cell proliferation, death and protein turnover.

The percentages and fluorescence intensities in B cells were determined by flow cytometry at different times after CFSE injection (as illustrated in Fig 4A). Upon injection, CFSE labeled similar proportions of cells (75%). The kinetics of CFSE labeling was similar in the non-B and B cell populations of sheep infected with wild-type and miRNA-deleted viruses (p = 0.31 and 0.47, respectively, according to non-linear mixed model) (Fig 4B and 4C). In B cells, CFSE labeling was significantly different only at day 23 (p = 0.02, according to t test).

Kinetic parameters were calculated with a mathematical model based on two data sets: "$I$" the ratio of the (mean intensity of fluorescence (MFI) of CFSE+ cells to the MFI of CFSE- cells and "$P$" the percentage of CFSE+ cells [32]. By fitting this model to the data, we were able to quantify two kinetic parameters: "$p$" (the average proliferation rate) and "$d$" (the average

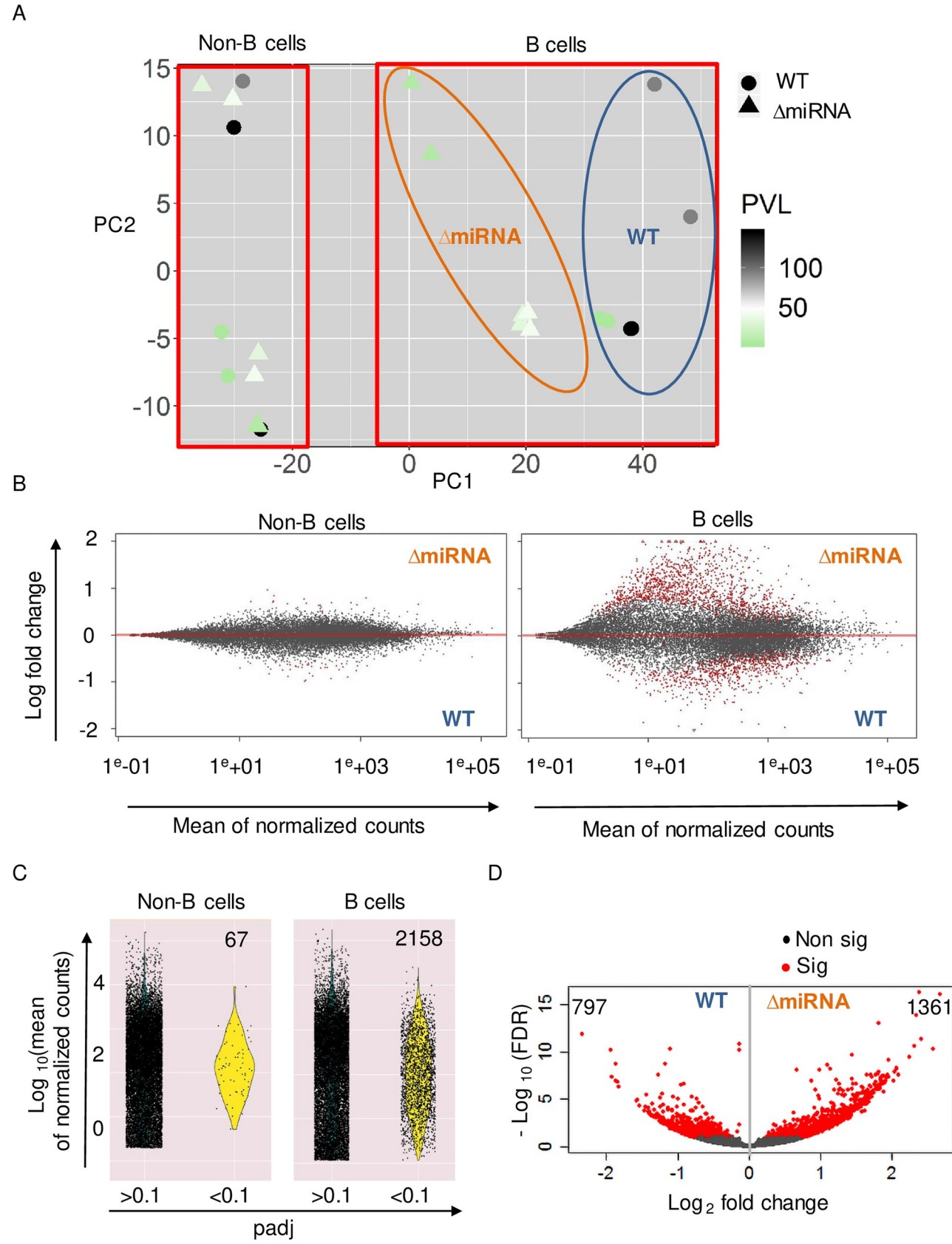

**Fig 2. Transcriptomic changes in cells isolated from sheep infected by wild-type and miRNA-deleted viruses.** (A) Principal component analysis of RNA sequencing data from B and non-B cells isolated from sheep infected with pBLV-ΔmiRNA (triangles) and pBLV-WT (circles). The proviral loads (PVL) are represented by a color code (from green to black). PCA was generated using DEseq2 and ggplot2 packages in R. (B) Plots of mean of normalized counts (x axis) and log fold change (y axis) in B and non-B cells. Dots are red if the adjusted p value (padj) is less than 0.1. (C) Distribution of the gene transcripts displayed in violin plots according to padj. (D) Volcano plot of differentially expressed genes in B cells of pBLV-ΔmiRNA and pBLV-WT-infected sheep. Data are plotted as $\log_2$ fold change (x axis) versus the $-\log_{10}$ of the false discovery rate (FDR) (y axis). Red dots represent significant genes with adjusted p value below 0.1.

disappearance rate). Compared to wild-type, the cell turnover of peripheral blood B cells was reduced in sheep infected with pBLV-ΔmiRNA (Fig 4D).

## DNA synthesis is reduced in B cells from pBLV-ΔmiRNA infected sheep

As illustrated by previous studies [11, 34], B cell proliferation occurs mostly in lymphoid organs. To complement CFSE experiments, we next evaluated incorporation of bromodeoxyuridine (BrdU) into the B cell nucleic acids. Upon intravenous injection of BrdU, the percentages of BrdU-positive B cells were determined by flow cytometry (Fig 5A). At day 1, the percentage of B cells having incorporated BrdU reached a maximum (2.4% ± 0.25) in sheep infected with wild-type virus (Fig 5B). In contrast, BrdU incorporation was significantly lower in sheep infected with pBLV-ΔmiRNA at day 1 (0.8% ± 0.3, p = 0.0006, according to t-test).

To quantify BrdU incorporation data, we used a mathematical model that considers (i) the rate of proliferation and death of the B and non-B cells, (ii) the loss of unincorporated label after injection and (iii) the dilution of the BrdU label consequent to cell division [35]. Fitting the mathematical model to the experimental data yielded significantly different average proliferation rates in sheep infected with pBLV-WT (4.9% ± 1.1) or with pBLV-ΔmiRNA (1.6% ± 0.8) (p = 0.02 according to Wilcoxon–Mann–Whitney test) (Fig 5C). The percentages of B cells that are generated by proliferation per day were thus approximately 3 times higher in wild-type infected animals. In contrast, no difference was observed in the non-B cell populations (p = 0.14 according to Wilcoxon–Mann–Whitney test).

Overall, these data demonstrate that the B cell turnover is decreased in sheep infected with the pBLV-ΔmiRNA deletant.

## Proliferation is reduced in spleen and lymph nodes from pBLV-ΔmiRNA infected sheep

Since B cell proliferation occurs in lymphoid organs, we analyzed Ki67 expression by immunohistochemistry in spleen and lymph node sections of pBLV-WT and pBLV-ΔmiRNA infected animals (Fig 6A and 6C, respectively). Label quantification showed that the average of Ki67 expression in spleen sections from pBLV-ΔmiRNA infected sheep (4.1% ± 0.3) was significantly lower compared to wild-type levels (9.8% ± 0.8; p<0.0001 according to t-test; Fig 6B). The percentages of Ki67 positive cells were also decreased in lymph nodes from infected sheep (3.2% ± 0.2 in pBLV-ΔmiRNA and 7.1% ± 0.5 in pBLV-WT infected sheep; p<0.0001 according to t-test; Fig 6D).

These data show that proliferation is reduced in lymphoid organs of sheep infected with the pBLV-ΔmiRNA deletant.

## Discussion

In this report, we have shown that BLV non-coding RNAs affect B-lymphocyte proliferation based on transcriptomic analyzes, in vivo kinetic data and immunochemistry. RNA sequencing provided a very clear view of their involvement in B cell proliferation. Indeed, twenty-two

A

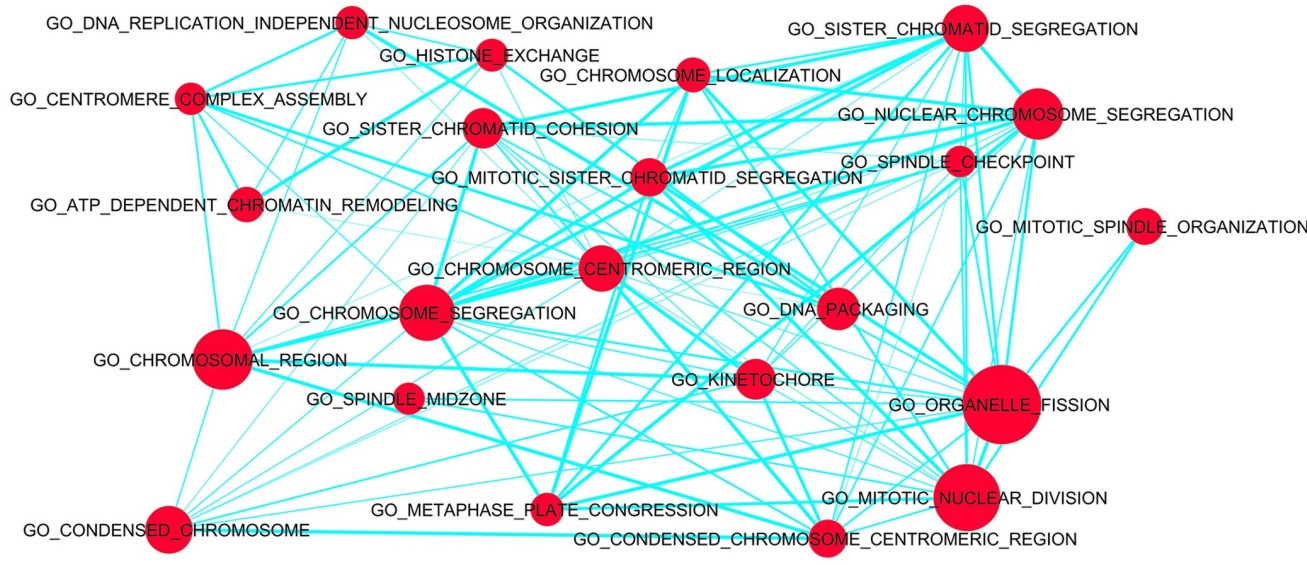

B

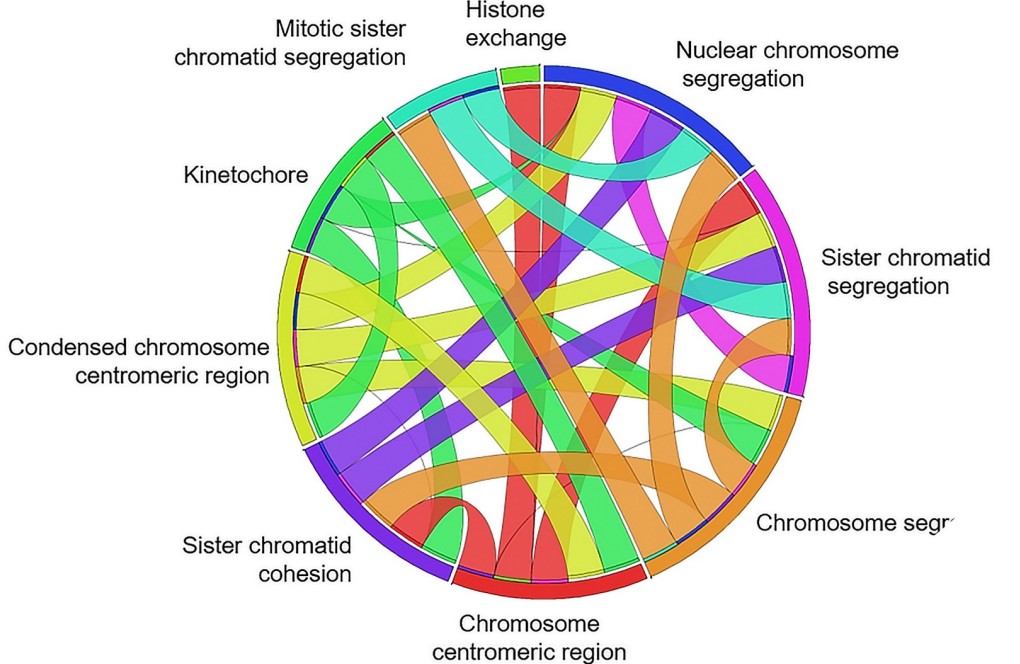

**Fig 3. Gene set enrichment in B cells from sheep infected by wild-type virus compared to miRNA deletant.** (A) Enrichment map of Gene Set Enrichment Analysis (GSEA) generated by Cytoscape. Displayed sets contain between 15 and 500 genes that are enriched with a false discovery rate less than 0.001 (FDR < 0.001). Red nodes symbolize enriched GO gene sets. Node size indicates the total number of genes in each gene set. Edge thickness (blue line) indicates the number of overlapping genes between gene sets computed based on Jaccard coefficient. (B) Chord diagram of enriched gene sets in B cells from pBLV-WT-infected sheep based on family wise-error rate less than 0.001 (FWER < 0.001). The segment represents the GO gene sets and the ribbon highlights the similarity between them.

A

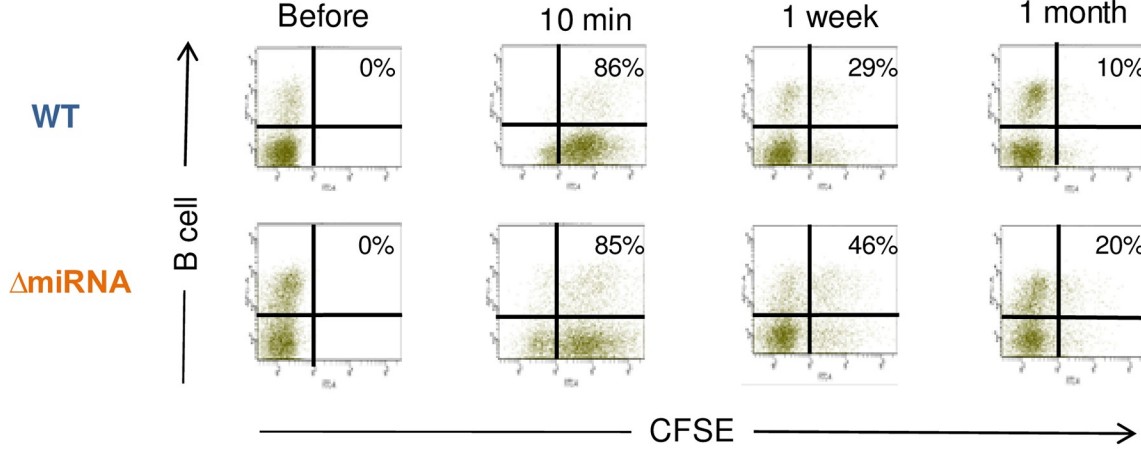

B                                                    C

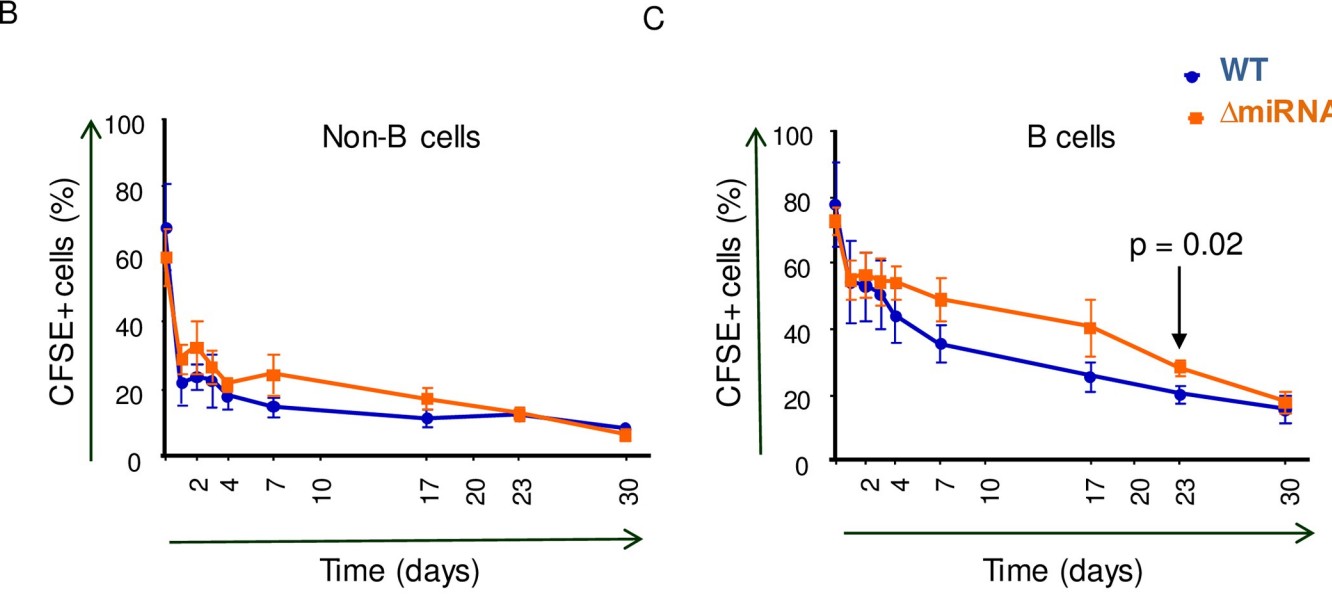

D

| Group | Proliferation rate | Death rate |
|---|---|---|
| WT | 0.081 ± 0.019 | 0.135 ± 0.022 |
| ΔmiRNA | 0.066 ± 0.018 | 0.092 ± 0.018 |

**Fig 4. CFSE kinetics in peripheral blood.** (A) Representative flow cytometry dot plots of CFSE-labeled B cells at different times after intravenous CFSE injection. Numbers represent the percentages of CFSE+ B cells within the total B-lymphocyte population. (B) Time kinetics of percentages of CFSE-labeled

non-B cells in total non-B cells. (C) Corresponding kinetics of the CFSE+ B cell population. p = 0.02 at day 23 according to t-test. All data are plotted as mean ± SEM (WT n = 3, ΔmiRNA n = 4). (D) Estimated B cell proliferation and death rates based on CFSE data modeling.

gene sets with false discovery rate less than 0.001 pertained to cell division. This observation is consistent with the paradigm of a virus that constantly attempts to proliferate under the control of a very efficient immune response [36]. Another outcome of RNA sequencing is that the transcriptome of T cells is not significantly affected in pBLV-ΔmiRNA infected sheep. For example, pathways of T-cell specific factors: CD4-specific cytokines (IL2; p = 0.6), T-reg (foxp3; p = 0.9), mediators of cytotoxic activity (granzyme; p = 0.7) are similar (S4 Fig). Since BLV infects exclusively B cells in the peripheral blood, it seems that non-coding RNAs from B cells are not transferred to T cells despite being highly expressed in the plasma. The biological role of plasmatic miRNAs, if any, is thus currently unknown. Possibly, the miRNAs only modulate the T cell response in specific tissues such as the lymph nodes of the spleen. Our data show that the BrdU kinetics of non-B cells is not significantly different in sheep infected with wild-type virus or with miRNA deletant. In fact, lack of major differences in the T cell transcriptome is concordant with similar T cell turnover rates.

Transcriptomic data in sheep thus revealed a very clear involvement of proliferation in B-lymphocytes without any major change in non-B cells. A similar approach with bovine PBMCs yielded a complex network of pathways modulated by the miRNAs including cell signaling, cancer genes and immune response [24]. In the ovine model, results from this report also highlight a role of BLV miRNAs in immune response modulatory pathways such as inflammation, leukocyte chemotaxis, cytokine secretion and receptor activity, adaptive and humoral immunity (S4 Table). Data also shows enrichment of specific signaling pathways including phosphatidylinositol 3 kinase, ERK1 and ERK2 and NF-kB in absence of miRNAs (S4 Table). Comparison between the two species is nevertheless difficult because, in our previous study in bovines, B cell populations were not sorted prior to RNA sequencing. It is therefore possible that a series of significant genes in B-lymphocytes were not identified due to the transcriptome of non-B cells. We analyzed the expression levels of genes that were previously identified as being controlled by the miRNAs in the bovine model. Among these, GZMA and PIK3CG are also significantly downregulated in the presence of BLV miRNAs in the ovine model (S5 Fig).

The gene ontology analysis (Fig 3) in B cells indicates a more specific effect of the miRNAs on mitosis than merely on proliferation. Most genes whose expression is increased in the presence of miRNAs indeed concern chromosome segregation at mitosis (S3 Fig and S2 Table). These include AURKB, the serine/threonine-protein kinase component of the chromosomal passenger complex (CPC), a complex that acts as a key regulator of mitosis [37]. BIRC5 directs CPC movement to different locations from the inner centromere during prometaphase to midbody during cytokinesis and participates in the organization of the center spindle by associating with polymerized microtubules [38]. Chromobox 5 (CBX5) involved in the formation of functional kinetochore through interaction with MIS12 complex proteins [39]. The serine/threonine-protein kinase BUB1 is essential for spindle-assembly checkpoint signaling and for correct chromosome alignment [40]. The BLV miRNAs thus affect expression of several genes that regulate spindle activity and chromosome segregation during mitosis.

Our interpretation of transcriptomic data was supported by in vivo measurement of cell kinetics. Two types of data sets were collected by flow cytometry: the rates of BrdU-incorporation into B cells and the percentages of B+CFSE+ cells. Both approaches are complementary: BrdU data inform about cell proliferation occurring in lymphoid organs and CFSE labeling provides the cell turnover of peripheral blood B-lymphocytes. Based on non-linear mixed

A

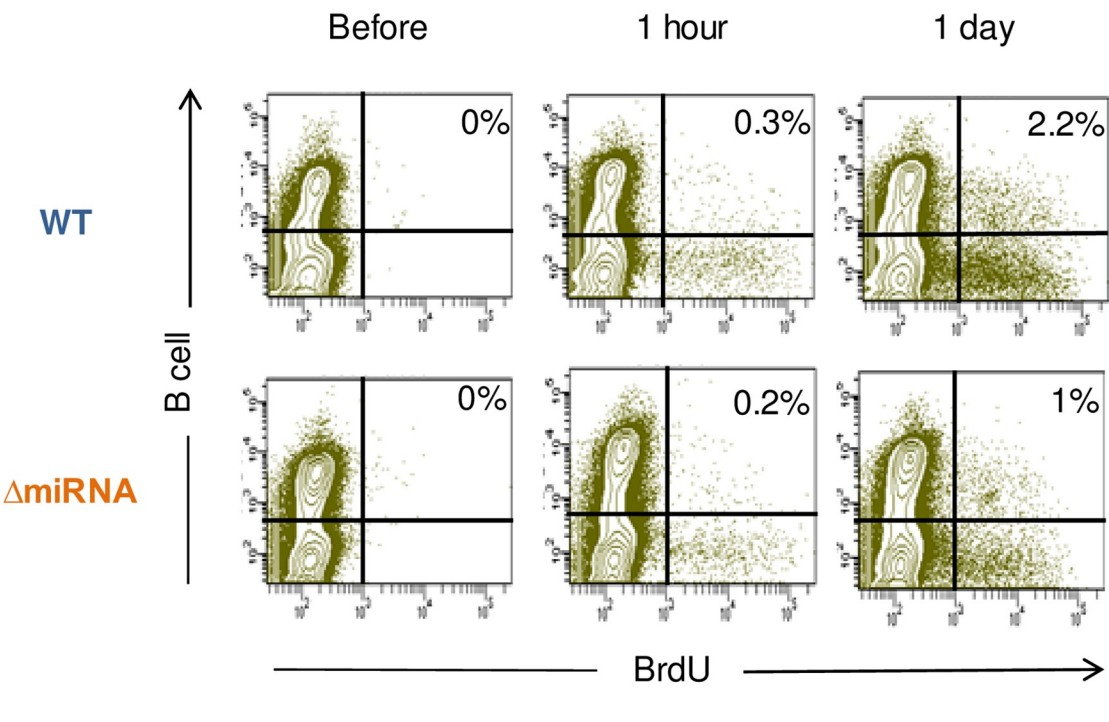

B

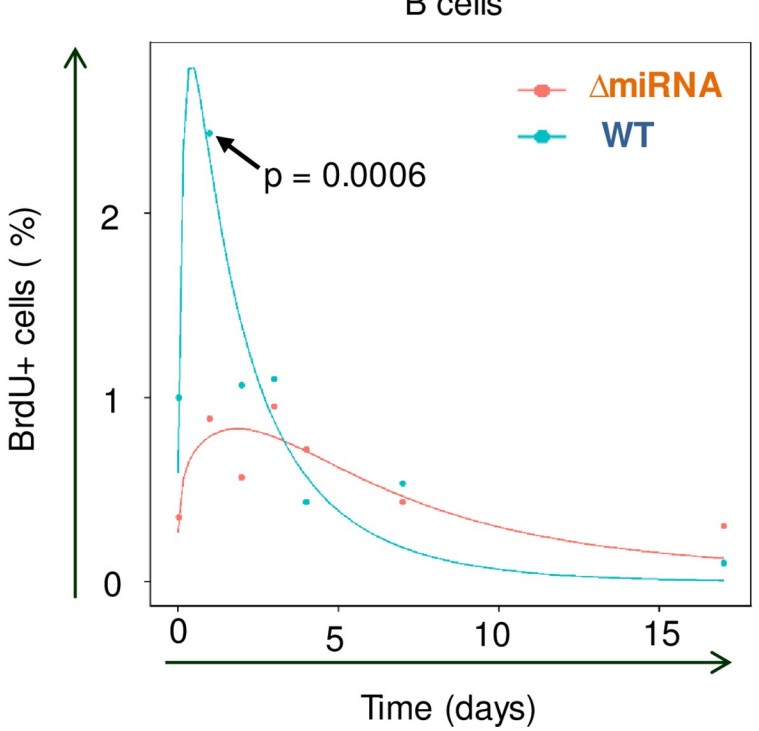

C

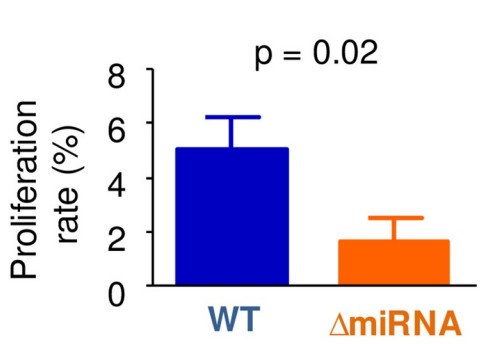

**Fig 5. Evaluation of proliferation rates by intravenous injection of BrdU.** (A) Representative flow cytometry dot plots of BrdU-labeled B cells at different times after intravenous BrdU injection. X axis corresponds to BrdU-FITC and y axis display B cells-APC. Numbers represent the percentages of BrdU-labeled B cells in the total B cell population. (B) Time kinetics of the percentages of B cells having incorporated BrdU. The arrow shows the percentage of B cells having incorporated BrdU at day 1 (p = 0.0006, according to t-test). (C) Estimated proliferation rates based on BrdU data; p = 0.02, according to Wilcoxon–Mann–Whitney test. Data are plotted as mean ± SEM (WT n = 3, ΔmiRNA n = 6).

models, Fig 4 reveals that the CFSE kinetics are similar in wild-type infected sheep and miRNA deletants. Since the value at day 23 is statistically different (Fig 4C), there is nevertheless a trend to a faster decrease in the presence of miRNAs that may be biologically relevant. Indeed, the two-fold increase in BrdU incorporation that occurs in lymphoid organs should be compensated by a higher death rate in the peripheral blood. Indeed, the total number of B-cells remained relatively constant during the experiment. We have previously published a model that reconciles these apparent discrepancies [36].

It should be mentioned that kinetic parameters were calculated for the total IgM-positive B cell population. In principle, it would be possible to individually determine the cell turnover of provirus-carrying cells by fluorescent in situ hybridization (FISH) coupled with BrdU labeling. Unfortunately, this technique yields some background that masks specific BrdU incorporation. It is noteworthy that deletant and wild-type infected animals with similar proviral loads still display differences in B cell proliferation (S6 Fig). It is thus not the level of proviral load by itself that explains the higher proliferation rate of B cells infected by wild-type virus. In any case, it is remarkable that proviral loads as little as 0.7 copies/100 PBMCs is still associated with a significant difference in cell proliferation (proliferation rate = 6.1% according to BrdU kinetics) [41, 42]. Notwithstanding, it remains unexpected that the B-cell proliferation rate differs between animals infected with wild-type and mutant virus even when the proviral loads are very similar (S6 Fig). This observation also implies that the risk of leukemia, which is significantly different (Fig 1B), might depend on the viral and/or host genotype independently of the proviral load. Analysis of the proviruses excluded that the miRNA deletion reverted to wild-type (e.g. by recombination with a virus from another sheep (S7 Fig). We were also unable to identify point mutations within the provirus that may support preferential replication. Although major histocompatibility complex class II DRB3 polymorphisms correlate with susceptibility for developing the disease [43], it is unlikely that variations in host genotypes systematically reproduce differences in B-cell kinetics (S6 Fig). Although there might be a difference in the clonal evolution of the infected cells, we favor a mechanism by which the miRNAs that are exported in the plasma affect proliferation of non-infected B cells. In fact, this mechanism possibly explains the increase of infected as well as non-infected B-cells during lymphocytosis (PL).

The best predictor of leukemia is the increase in the proviral load, followed by an inversion of the B/T ratio [44]. Our data indicate that the miRNAs contribute to oncogenesis by promoting B-cell proliferation. In contrast, the T cell response does not correlate with the proviral loads, suggesting that their stimulation by viral antigens is not a limiting step. Long-term follow-up of sheep infected with the ΔmiRNA mutant reveals that the microRNAs are required for pathogenesis in sheep (Fig 1). There was one wild-type infected sheep with slow rise in proviral load (Fig 1A), suggesting that individual variations between host's genotype may affect viral replication. Despite a long term follow up, we can also not exclude that oncogenesis will never occur in the absence of miRNAs. Indeed, one sheep (#1131) infected with a ΔmiRNA mutant currently carries high BLV proviral loads (S7 Fig). With an estimated B-cell proliferation rate of 0.08%, this sheep was infected with a ΔmiRNA deletant as illustrated in panel C of S7 Fig. In fact, we have previously experienced a similar situation with another BLV mutant devoid of R3-G4 accessory genes: only 1/20 developed leukemia/lymphoma 6 years post-inoculation [45]. Whether the miRNAs are directly involved in oncogenesis will require further

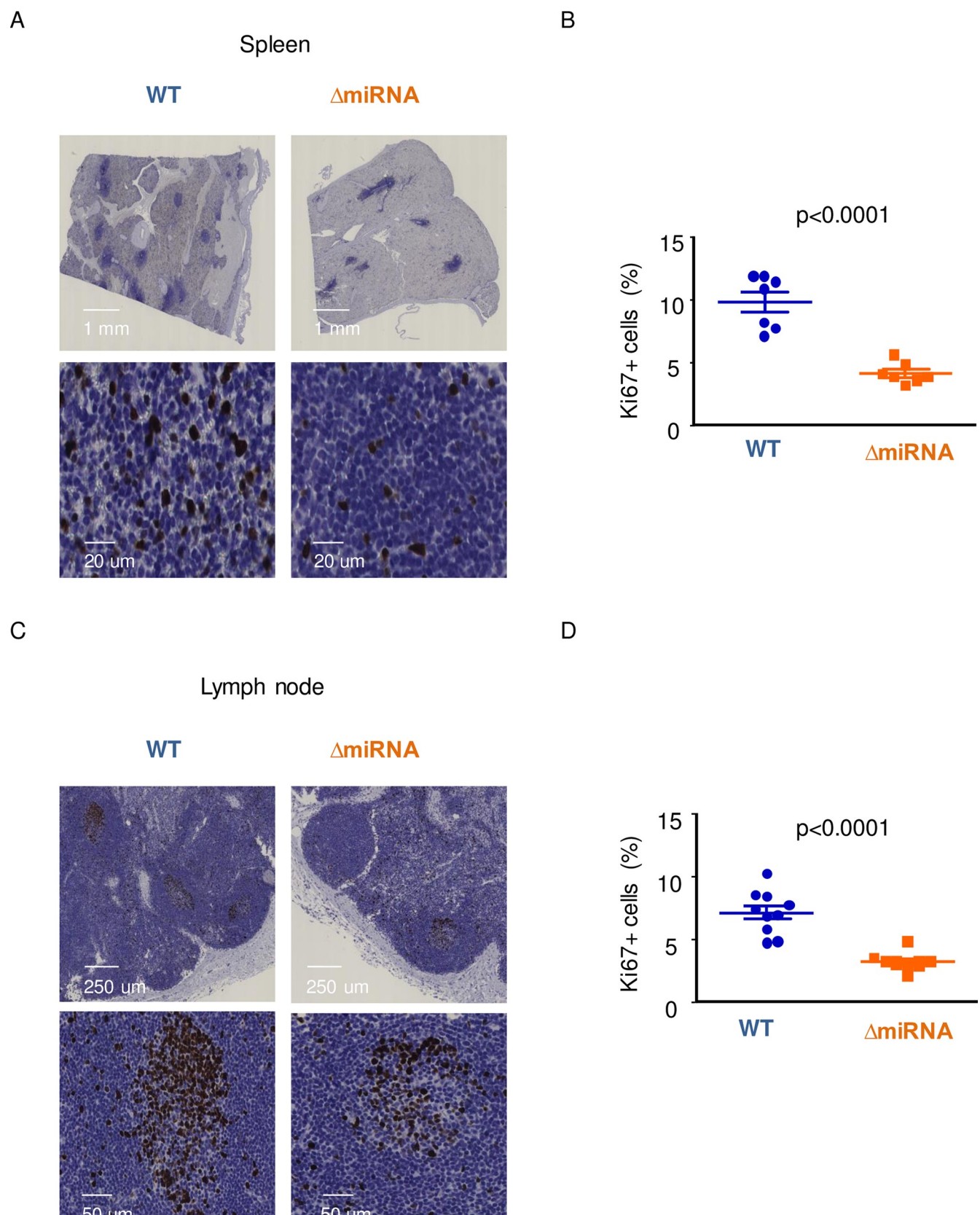

**Fig 6. Assessment of proliferation rates in spleen and lymph nodes.** Immunohistochemistry analysis of Ki67 in spleen (A) and in lymph node (C). Quantification of Ki67-positive cells in sections of spleen (B) (n = 7) and in lymph nodes (D) (n = 11).

follow-up of a larger number of sheep. It is noteworthy that this question is extremely difficult to address in the bovine species considering the long latency period (7–10 years) and the low frequency of leukemia/lymphoma (5–10%). Providing that, among 6 sheep, #1131 further progresses to leukemia and considering a frequency ratio of 13.33-fold (i.e. 100% in sheep / 7.5% in cows), we estimate that a follow up of 80 bovines during 8.5 years (7–10) are needed to determine whether miRNAs are required for oncogenicity in this species.

Strength of the reverse genetics approach is that the role of genetic determinants can be investigated in relevant conditions in vivo (i.e. in the context of a provirus expressed at physiological levels in the correct cell type). A potential risk is that other steps in the viral life cycle may be affected (e.g. reverse transcription, integration, infectivity, packaging or budding). We think that this possibility is unlikely because no major difference was observed at seroconversion indicating that essential steps of the viral life cycle are unaffected. Similarly, deletion of the miRNAs also affects the sequence of the antisense transcripts (AS1L and AS1S RNAs but not AS2). Specific mutations of these non-coding transcripts will be required to address this issue. In this context, it is peculiar that human T-cell leukemia virus type 1 (HTLV-1) closely related to BLV has no similar cluster of RNAPIII-driven miRNAs. As BLV AS1 and AS2, HTLV-1 nevertheless encodes a poorly translated antisense RNA (HBZ) that mostly remains in the nucleus. When the methionine initiation codon is mutated, untranslatable HBZ RNA promotes cell proliferation [46]. In this perspective, it appears that both BLV and HTLV-1 have developed strategies to affect cell proliferation using non-coding or at least poorly translated ribonucleic acids.

Other viruses, such as SFV, Marek's disease virus type 1 (MDV), human cytomegalovirus (CMV), Kaposi's sarcoma-associated herpesvirus (KSHV) and Epstein–Barr (EBV) express non-coding RNAs to modulate cell fate [47, 48]. KSHV encodes miRNAs to modulate replication and pathogenesis in B cell hyperproliferative disorders, including Primary Effusion Lymphoma (PEL) and some forms of Multicentric Castleman's Disease (MCD) [49, 50]. Similarly, EBV miRNAs promote cell survival and proliferation during latent infection [51]. Sharing a seed region with a host miRNA enables virus-encoded miRNAs to modulate specific functions [47]. For example, the cellular miR-155 is mimicked by pro-oncogenic miRNAs encoded by KSHV and MDV [52, 53]. Ablation of these viral miRNAs abrogates KSHV and MDV induced lymphoma [54]. Two SFV miRNAs mimics seed sequence and function of the host lymphoproliferative miRNA, miR-155 and the innate immunity suppressor miR-132 [48]. Likewise, BLV encodes a miRNA (mir-B4) an analog of cellular miR-29 that is involved in multiple oncogenic processes, including proliferation [18].

Together, our data reveal that the major function of the BLV miRNAs is to promote cell proliferation. In fact, viral replication via the infectious cycle (i.e. expression of viral particles and infection of new cells) is extremely inefficient after that the immune response is established [12]. Soon after seroconversion, the mode of viral replication switches to clonal expansion of provirus-carrying cells. This mode of replication requires that infected cells undergo mitosis more frequently than other B-lymphocytes. Alternatively, or concomitantly, infected cell clones may also expand providing that the death rates are reduced [36]. Our data of Fig 5 show that 4.9% B cells from wild-type virus infected sheep are produced by proliferation per day. This accelerated proliferation is only partly compensated by increased death. The disequilibrium between cell death and proliferation rates is predicted to enforce progressive accumulation of B cells in animals infected by wild-type virus.

In conclusion, combined bioinformatics and in vivo kinetic experiments reveal that BLV non-coding RNAs primarily promote cell proliferation of BLV-infected lymphocytes. Ablation of non-coding RNAs results in reduced proliferation of the infected cell and delayed clonal expansion as illustrated by the kinetics of proviral loads. Ultimately, absence of miRNAs is also associated with lack of pathogenesis.

## Materials and methods

### Ethics statement

All animal studies were conducted with the approval of the Institutional Committee for Care and Use of Experimental Animals under protocol number 1515. Sheep were kept under controlled condition at the animal facility CEPA of the university.

### Virus inoculation into sheep and quantification of proviral loads

Construction of proviral vectors (pBLV-ΔmiRNA and pBLV-WT) and inoculation protocols were described elsewhere [24]. pBLV-ΔmiRNA is isogenic to pBLV-WT but contains a deletion of the miRNA coding region (nucleotides 6170 to 6736 according to the BLV reference genome NC_001414.1) [24]. At regular intervals of time, blood was collected by jugular venipuncture. PBMCs were isolated by Percoll density gradient centrifugation, frozen in FBS containing 10% dimethyl sulfoxide (DMSO) (Sigma-Aldrich) and kept at -80˚C or liquid nitrogen.

To quantify the proviral loads, genomic DNA was extracted from PBMCs using DNeasy Blood and Tissue kit (Qiagen) following manufacturer's recommendations. 100ng of genomic DNA was used for real-time PCR amplification of BLV proviral sequences as described previously [24]. Proviral loads were determined from 3 independent qPCR amplifications of DNAs extracted independently.

### RNA sequencing of B and non-B cell populations

To determine the percentages of B cells, PBMCs were washed twice in PBS supplemented with 10% fetal bovine serum (FBS) and labeled with an anti-IgM monoclonal antibody (clone 1H4, 1:100 dilution of hybridoma supernatant) for 30 minutes at 4˚C. After two washes, cells were incubated with Alexafluor 647 goat anti-mouse IgG1 conjugate (Thermo fisher scientific, 1:1000 dilution) and analyzed with a FACS Aria (Becton Dickinson).

B cells were purified from the PBMCs using MACS positive selection LS columns (Miltenyi Biotec). Briefly, freshly isolated PMBCs were labeled with 1H4 antibody, washed and incubated with goat anti-mouse IgG microbeads (Miltenyi Biotec). Each sample was magnetically sorted at room temperature using LS columns inserted into a QuadroMACS™ separator. To increase the purity of the magnetically labeled fraction, the eluted fraction was enriched by a second round of MACS. The purity of MACS-sorted cells (> 95%) was verified by flow cytometry.

RNA was isolated from MACS-separated B and non-B cells using the miRNeasy Mini Kit (Qiagen) following manufacturer's protocol after removal of contaminating DNA (Thermo Fisher Scientific). After determination of their concentration (Quant-IT RiboGreen, Invitrogen), samples were run on TapeStation RNA screentape (Agilent). Only high-quality RNA preparations, with RNA integrity number (RIN) greater than 7.0, were used for RNA library construction. Libraries were prepared with 1µg of total RNA using the Illumina TruSeq stranded mRNA Sample Prep kit (Illumina). The libraries were quantified using the KAPA Library Quantificatoin kit for Illumina Sequencing platforms and qualified using the

TapeStation D1000 ScreenTape (Agilent Technologies). Indexed libraries were then sequenced using the NovaSeq 6000 platform (Illumina).

## Bioinformatics

FastQC software was used for quality control, visualization, and quantification of raw data. Aligning and mapping the raw sequencing data to the sheep reference genome (Oar_v3.1) was performed by STAR v2.4.0.1 [55]. FeatureCounts was used for read quantification [56]. Normalization and differential gene analysis was performed with R packages DESeq2 [27]. For the identification of enriched transcriptomic signatures, differential gene expression list was loaded on the gene set enrichment analysis (GSEA) tool (v3.0) from the Broad Institute at MIT [29]. We used C5: GO gene sets from MSigDB to interpret the transcriptomic signatures. Leading edge analysis on enriched gene sets was analyzed using GSEA.

## Quantification of peripheral blood B cell turnover in vivo

Twenty-five mg of 5(6)-carboxyfluorescein diacetate N-succinimidyl ester (Sanbio) dissolved in 4 ml of dimethyl sulfoxide and mixed with 1,000 U/ml heparin sodium salt (Santa Cruz Biotechnology) were injected into the jugular vein of sheep as described before [10]. At regular time intervals, blood was collected by jugular venipuncture. After PBMC separation, CFSE labeling of B cells was determined by flow cytometry using 1H4 monoclonal antibody and Alexa fluor 647 goat anti-mouse IgG conjugate (Thermo fisher scientific, 1: 1000 dilution). Proliferation $"p"$ and death $"d"$ rates were determined according to a model described in reference [32]. In brief, we considered that CFSE labeling halved upon mitosis since the dye was distributed in each daughter cell. The model uses two pieces of data from the flow cytometry analyses: the proportion of CFSE+ cells $"P"$ and the ratio of the mean fluorescence intensity of the CFSE+ population to the CFSE–population $"I"$ to estimate the rate of proliferation and the rate of death of CFSE labeled B lymphocytes. The cell populations undergoing five divisions are $x_0 = —(p + d)x_0$; $x_1 = 2px_0 - (p + d)x_1$; $x_2 = 2px_1 - (p + d)x_2$; $x_3 = 2px_2 - (p + d)x_3$; $x_4 = 2px_3 - (p + d)x_4$; $x_5 = 2px_4 - (p + d)x_5 + \lambda$ ($x_i$ being the proportion of B cells that have undergone i divisions since CFSE labeling). In the model, the cells in the $x_5$ category are CFSE–(either because they have divided 4 to 6 times since labeling and therefore lost their fluorescence or because they were not labeled by the initial injection). The average proliferation rate of cells is $"p"$, the average disappearance rate is $"d"$ and the average replacement rate is $\lambda$. These equations were solved analytically and then used to find expressions for $"I"$, the ratio of the mean fluorescence intensity (MFI) of the CFSE+ population to the CFSE–population, and $"P"$ the proportion of CFSE+ cells. For five divisions, the relevant equations are as follows:

$$I = ((Jx_0 + J/2x_1 + J/4x_2 + J/8x_3 + J/16x_4)/(x_0 + x_1 + x_2 + x_3 + x_4))/((J/32x_5)/(x_5))$$
$$= (4(24 + 24pt + 12p^2t^2 + 4p^3t^3 + p^4t^4))/(3 + 6pt + 6p^2t^2 + 4p^3t^3 + 2p^4t^4)$$

$$P = (x_0 + x_1 + x_2 + x_3 + x_4)/(x_0 + x_1 + x_2 + x_3 + x_4 + x_5)$$
$$= Fe^{-(p+d)t}(1 + 2pt + 2p^2t^2 + 4/3^3t^3 + 2/3p^4t^4)$$

$J$ is the MFI of CFSE label in undivided cells and $F$ is the proportion of peripheral blood B cells labeled by the initial injection. These formulas were fitted to the experimental data by resolving the non linear equation system (R packages systemfit v1.1–22 and nlstools' v1.0–2) after estimating the adequate number of divisions to reach negative status and the lymphocyte kinetics parameters estimated [57, 58].

### Analysis of 5-bromo-2-deoxyuridine incorporation in vivo

Sheep were injected intravenously with a single dose of 400 mg BrdU (Sigma) dissolved in 5 ml 0.9% NaCl. Serial analysis at different times (1 hour and 1, 2, 3, 4,7 and 17 days) was done without the reinjection of BrdU. To evaluate BrdU incorporation into B lymphocytes, PBMCs were isolated and labeled with PIG45A monoclonal antibody (A&E Scientific) for 30 min at 4˚C and, after two washes, revealed with Alexa Fluor 647 goat anti-mouse IgG2b conjugate (Thermo fisher scientific, 1: 1000 dilution). Then, cells were fixed, permeabilized and treated with DNase using BrdU flow Kit reagents according manufacturer's protocol (BD Biosciences). Finally, the cells were labeled by FITC-conjugated anti-BrdU antibody (1:50 dilution) and analyzed by flow cytometry using a FACS Aria (Becton Dickinson).

Estimation of proliferation and death rates was done as previously described. In summary, we used following differential formula to the BrdU incorporation data achieved experimentally $dl/dt = 2\sigma pu + pl - dl$ where $u$ is the proportion of unlabeled cells and $l$ is the proportion of labeled cells, "$p$" presents the average proliferation rate of cells, and "$d$" gives the average death rate of labeled cells. $\sigma$ is the probability that a proliferating cell becomes labeled. The probability that a proliferating B cell converts labeled is assumed to be an exponentially declining by time, $\sigma = e^{-\alpha t}$, reflecting the loss of unincorporated BrdU from the cytoplasm of cells. The differential model was fitted to the experimental data using differential equation adjustment (R package deSolve v 1.21) and the lymphocyte kinetics parameters estimated [59].

### Immunohistochemistry of spleen and lymph node biopsies

Spleen and lymph nodes were fixed overnight at room temperature in PBS containing 4% formaldehyde and stored in 70% ethanol. Immunohistochemistry with antibodies directed against Ki67 was performed by the GIGA immunohistology platform using established protocols [60]. Briefly, tissue sections were subjected to heat-induced epitope retrieval using a pressure cooker, rinsed in water and incubated in 3% hydrogen peroxide in methanol for 30 min. After washing in PBS, non-specific binding was reduced by incubation with normal goat serum. Then, samples were labeled with anti-Ki67 antibody (# 790–4286, Roche) for 1 hour at room temperature, washed twice with PBS and incubated with an anti-rabbit peroxidase conjugate (# K4003, Dako) for 30 minutes. Samples were revealed with diaminobenzidine tetrahydrochloride (DAB), washed with distilled water and observed under light microscopy with a 40× objective. Quantification of scanned images was performed with QuPath (0.1.2).

### Statistics

Statistical tests were performed using R v3.6.0 or GraphPad Prism 5. The t-test was used for statistical evaluations of proviral loads. Kaplan-Meier survival curves were compared by the Log-rank Mantel-Cox test [61]. CFSE and BrdU kinetics were assessed by daywise paired t-test. Proliferation and death rates were compared between groups by Wilcoxon-Mann-Whitney test.

## Supporting information

**S1 Fig.** (A) PBMCs were isolated from sheep infected by wild-type BLV and miRNA deletant. Then, PBMCs were labeled with anti-IgM 1H4 antibody and anti-mouse IgG1 conjugate. Percentages of B cells were determined by flow cytometry. p = 0.002 according to Mann-Whitney U test. (B) After overnight culture of PBMCs, the percentages of p24-positive cells were determined by flow cytometry. The p24 viral protein was detected by sequential incubation with 4′G9 monoclonal antibody and a rat anti-mouse IgG1 conjugate. p = 0.01 according to Mann-

Whitney U test. (C) Absolute number of lymphocytes determined with a ProCyte Dx Haematology Analyser. (D) Absolute numbers of B cells were calculated from the percentages of B-lymphocytes in PBMCs (panel A) and absolute numbers of lymphocytes (panel C).
(TIF)

**S2 Fig. Enrichment plot of gene sets with the family wise error rate less than 0.001.** The green curve corresponds to the ES (enrichment score) curve, which is the running sum of the weighted enrichment score obtained from GSEA software. The enrichment score reveals the degree at which the genes in a gene set are overrepresented at the top or bottom of the entire ranked list of genes (y axis).
(TIF)

**S3 Fig. Leading genes of the most enriched gene sets.** Chord diagram displaying leading edge analysis of enriched gene sets (FWER < 0.001) in pBLV-WT-infected sheep analyzed by GSEA. The diagram was generated by circos table viewer. Segments size shows the contribution effect.
(TIF)

**S4 Fig. Normalized transcriptomic counts of T-cell specific factors.** Normalized counts were obtained by DEseq2 analysis of transcriptomic data of non-B cells isolated from pBLV-WT and pBLV-ΔmiRNA infected sheep. Differences of gene expression between pBLV-WT and pBLV-ΔmiRNA are not significant according to t-test.
(TIF)

**S5 Fig. Normalized transcriptomic counts of GZMA, PPT1, FOS, ANXA1, MAP2K1 and PIK3CG.** (A) Normalized counts obtained from DEseq2 analysis of transcriptomic data of non-B cells isolated from pBLV-WT and pBLV-ΔmiRNA infected sheep. Differences of gene expression between pBLV-WT and pBLV-ΔmiRNA are not significant according to t-test. (B) Normalized counts obtained from DEseq2 analysis of B cells. Differences are significant for GZMA (p = 0.007) and PIK3CG (p = 0.02) according to t-test.
(TIF)

**S6 Fig. Evaluation of proliferation rates by intravenous injection of BrdU in animals with similar proviral loads.** (A) Time kinetics of the percentages of B cells having incorporated BrdU. (B) Proviral loads (in number of copies in 100 PBMCs) and proliferation rates corresponding to graphs of panel A.
(TIF)

**S7 Fig. BrdU kinetics in preleukemic sheep #1131.** (A) Time kinetics of the percentages of B cells having incorporated BrdU in animal # 1131 infected with pBLV-ΔmiRNA (B) Proliferation rate estimated from data of panel A. (C) PCR amplification of the genomic sequences surrounding the miRNA region. (D) Kinetics of proviral loads (in number of copies in 100 PBMCs) in sheep #1131.
(TIF)

**S1 Table. Differentially expressed genes that are common to B cells and non-B cells.** Genes significantly differentially expressed in B cells were compared to genes significantly differentially expressed in non-B cells. The table shows the genes that are shared by these two lists.
(XLSX)

**S2 Table. Leading genes of upregulated pathways in B cells of pBLV-WT infected sheep as compared to pBLV-ΔmiRNA.** Genes driving the enrichment score (Fig 3B) were identified by leading edge (LE) analysis on enriched gene sets with family wise-error rate <0.001 using the

GSEA software. The list of the genes has been ordered according to $\log_2$ fold change.
(XLSX)

**S3 Table. Upregulated pathways in B cells of pBLV-WT infected sheep as compared to pBLV-ΔmiRNA.** Gene ontology sets that are enriched in B cells of pBLV-WT infected sheep with a false discovery rate less than 0.01 (FDR < 0.01) were calculated using GSEA and listed according to the family wise-error rates (FWER p value). The size indicates the number of genes in each GO. Enrichment Score (ES) is the degree at which the genes in a gene set are overrepresented at the top or bottom of the entire ranked list of genes. NOM p values are the normalized p values calculated by GSEA. FDR q values represent false discovery rates.
(XLSX)

**S4 Table. Upregulated pathways in B cells of pBLV-ΔmiRNA infected sheep as compared to pBLV-WT.** Gene ontology sets that are enriched in B cells of pBLV-ΔmiRNA infected sheep with a false discovery rate less than 0.01 (FDR < 0.01) were calculated as described in S3 Table.
(XLSX)

## Acknowledgments

We thank the GIGA technological platforms in particular the genotranscriptomic and immunohistology facilities. We are grateful to Becca Asquith, Catherine Charles and Benoît Charloteaux for advice and expertise in modeling and bioinformatics.

## Author Contributions

**Conceptualization:** Roghaiyeh Safari, Luc Willems.

**Data curation:** Roghaiyeh Safari.

**Formal analysis:** Roghaiyeh Safari, Jean-Rock Jacques, Yves Brostaux.

**Funding acquisition:** Luc Willems.

**Investigation:** Roghaiyeh Safari, Jean-Rock Jacques, Yves Brostaux, Luc Willems.

**Methodology:** Roghaiyeh Safari, Jean-Rock Jacques, Yves Brostaux, Luc Willems.

**Project administration:** Luc Willems.

**Resources:** Luc Willems.

**Software:** Roghaiyeh Safari.

**Supervision:** Luc Willems.

**Validation:** Roghaiyeh Safari, Luc Willems.

**Visualization:** Roghaiyeh Safari.

**Writing – original draft:** Roghaiyeh Safari.

**Writing – review & editing:** Roghaiyeh Safari, Luc Willems.

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
