## [Decision Letter · Decision Letter 0]

17 Feb 2020

Dear Pr Willems,

Thank you very much for submitting your manuscript "Ablation of non-coding RNAs affects bovine leukemia virus B lymphocyte proliferation and abrogates oncogenesis" for consideration at PLOS Pathogens. As with all papers reviewed by the journal, your manuscript was reviewed by members of the editorial board and by several independent reviewers. While reviewer 1 was enthusiastic, both reviewer2 and reviewer 3 felt that the results presented were somewhat predictable in light of earlier work from the same group but could be strengthened by, for example, more mechanistic insights. In light of the reviews (below this email), we would like to invite the resubmission of a significantly-revised version that takes into account the reviewers' comments. This is subject to re-review by the same reviewers.

We cannot make any decision about publication until we have seen the revised manuscript and your response to the reviewers' comments. Your revised manuscript is also likely to be sent to reviewers for further evaluation.

Sincerely,

Bryan R. Cullen

Associate Editor

PLOS Pathogens

Susan Ross

Section Editor

PLOS Pathogens

Kasturi Haldar

Editor-in-Chief

PLOS Pathogens

orcid.org/0000-0001-5065-158X

Michael Malim

Editor-in-Chief

PLOS Pathogens

orcid.org/0000-0002-7699-2064

Reviewer's Responses to Questions

**Part I - Summary**

Reviewer #1: It has been recognized for some years that bovine leukemia virus (BLV) encodes several RNAs that do not encode proteins, but the functions of these non-coding RNAs are not completely understood. In this paper, the authors use the well established model of infection of sheep with BLV, to compare the wildtype virus with the same strain from which the non-coding RNAs have been knocked out. They conclude that an important function of the non-coding RNAs is to promote proliferation of BLV-infected B cells in vivo. The paper is well written, the data are clearly presented and for the most part support the major conclusions drawn by the authors.

Reviewer #2: Safari et al describe the role of BLV non-coding RNAs on viral replication, leukemia development and regulation of host cell gene expression in sheep using a reverse genetic approach.

The same group has previously performed a very similar study (published in 2016) in a bovine model showing that the viral miRNAs regulate the expression of genes involved in cell signaling, cancer and immunity. As the approach used in the two studies is the same, the novelty of the data described in the current manuscript is limited. Although the sheep model allows to evaluate the long-term effects of the viral miRNAs compared to the bovine animal model, no mechanistic analyses have been included, thus limiting the impact of the new data compared to previous observations.

Reviewer #3: In this manuscript Safari et al described the role of non coding RNA in the B-cell biology after experimental infection of sheep animals with BLV. They confirmed their previous observation, showing that sheeps infected with BVL devoided of miRNA have a reduced viral load compare to infection with wt virus and survive to the infcetion while sheeps infected with virus didn't.

In this study, the deeply analyzed the role of miRNA in Bcell compartment. Using transcriptomic analysis of B-cells purified from PBMCs of sheeps infected with miRNA deletant virus or wt virus, they identified that viral miRNA mainly target B-cell and not non B-cells population and mainly genes involved in proliferation pathways. Using, in vivo labeling they further demonstrated that B-cells from wt infected animals had a higher proliferation rate than B-cells from miRNA deletant infected animals.

The use of in vivo infection and analysis of freshly purified cells is certainly an important strength of this study.

The main weaknesses are linked to the lack of mechanisms supporting the lower B-cell proliferation rate after infection with miRNA deletant. Specially, since the author have reported a role of miRNA in controlling expression of gene involved in cell signaling, cancer and immunity, it would have been interested to address whether these gene are involved in B-cell proliferation and or also upregulated in B cell from animals infected with miRNA deletant.

In addition it would be interested to determine whether the 2x increase in B cells proliferation observed in wt infected animals translated to higher B cells total count reminiscent to oncogenesis.

**Part II – Major Issues: Key Experiments Required for Acceptance**

Reviewer #1: 1. Throughout the paper, the emphasis presented by the authors is on the impact on proliferation of knocking out the non-coding RNAs. However, the gene ontology analysis (results summarized in Figure 3) appears to suggest a more specific effect than merely on proliferation. Most of the genes whose expression is altered in the B cell infected by the knockout virus are concerned with the mechanics of chromosome segregation at mitosis. Obviously, these changes will be associated with a change in the cell proliferation rate. But these results suggest that the microRNAs might be exerting an impact on one particular stage of mitosis, and might even suggest some candidate targets of the non-coding RNAs. The authors might comment on this, if they agree, in the Discussion. It would be very interesting to pursue this in future work.

2. Lines 254-261. It is a curious observation that the B-cell proliferation rate differed between the two groups of animals even when the proviral loads were very similar. This observation also implies that the risk of disease (leukemia), which differed between the groups (cf. survival curves, Fig. 1B), might depend on the viral genotype independently of the proviral load. In turn this raises the question whether there is a difference between the groups in the clonal composition of what is presumably a highly polyclonal infected B-cell population, i.e. whether there is selection for particular different BLV-infected B-cell clones, which differ in leukemogenic potential. Have the authors studied the clonality of these populations?

3. Line 175 and fig. 4C. It is incorrect to use the t-test (whether paired or not) to test for a possible difference between the two curves in Fig. 4C. The reason for this is that the different timepoints in each respective curve are not independent of each other: that is, the percentage of CFSE+ cells at a given timepoint is not independent of the value at the previous timepoint on that curve. One can legitimately compare the two curves at each timepoint individually, and then correct for 9 multiple comparisons (9 timepoints were examined): clearly the difference would be significant only at some timepoints, if any.

Similarly, one cannot legitimately use regression techniques to test for a difference in the location of the two curves, again because of the non-independence of the plotted points.

The analysis, interpretation and conclusions should be revised to take this into account.

Reviewer #2: Authors have identified cell division as the major pathway regulated by BLV miRNAs in infected sheep. However, authors do not provide a detailed analysis of potential key genes in this pathway which could be directly affected by miRNAs and that could explain the different phenotype caused by the WT virus and the ΔmiRNA virus. What other pathways, besides cell division, are affected by the absence/presence of viral miRNAs?

Authors mention in the discussion that in bovine PBMCs the miRNAs modulated a more complex network of pathways, but they do not discuss a more detailed comparison of the results observed in the two animal models, which would instead be interesting, e.g. is the gene expression profile in bovine and ovine cells very different? Are the pathways identified in the bovine model (cell signaling, cancer and immunity) also affected in sheep?

• Fig 4: why the 2-fold increase shown in panel A at 1 month is not reflected in panel C?

• In panels C and D, authors state that viral miRNAs significantly control B cell turnover, cell proliferation and death, but the percentage of CFSE+ cells, proliferation rate and death rate are very similar between WT and ΔmiRNA viruses, with a fold difference less than 2. It’s not clear how biologically relevant this difference is. The authors should comment on this point?

• Fig 5A:

- At what point of the study was the BrdU analysis done?

- In the material and methods section, authors state that PBMCs were isolated at different time points after BrdU injection. It would be interesting to see the BrdU incorporation data at later time points (day 2-17): does the difference between WT and ΔmiRNA viruses become more pronounced?

Reviewer #3: 1- Using the transcriptomic data set what is the B-cell and non B-cell expression of ANXA1, FOS, GZMA, PIK3CG and PPT1 that were previously shown to be controlled in PBMCs by BLV miRNA ? And what is the link of these genes with B cell proliferation ?

2- Transcriptomic pinpoint to B-cell proliferation that was confirmed by CSFE and BrdU labelling. However the total count of B-cell were not provided in these experiments. What is the total number of B-cell in animals infected with miRNA deletant compare to animals infected with wt ? Does this number correlate with the 2X increase in proliferation that was demonstrated in fig 4 and 5 ? Is B-cell proliferation also observed after in vitro culture of purified B-cells ? ARe in vitro infected B-cells with miRNA deletant also impaired in their proliferation ability compared to wt in vitro infected B-cells ?

3- does the B cell proliferation correlate with the PVL ? Specially B-cell proliferation from animals with similar low PVL after infection with miRNA deletant or wt virus should be shown (p 12 lane 254-255

4- are the B cell that proliferate infected ? since PVL and infection are lower in miRNA deletant infected animals, one could expect that lower infected cells would translate in lower proliferation rate as measured in bulk B-cell population.

**Part III – Minor Issues: Editorial and Data Presentation Modifications**

Reviewer #1: 1. Figure 1. Can the authors comment on the discrepant results in the WT animal with the slow rise in PVL?

2. line 123. The authors write that ‘There was a clear segregation of the two principal components (PC1 and PC2)...’ But in Figure 2A, PC2 appears to differ between WT and ΔmiRNA animals, whereas PC1 does not.

3. line 131. Please expand the abbreviations MA and LFC.

4. lines 238-244. Since the previous study by this group quantified RNA expression in mixed, unsorted PBMCs, it is very difficult to compare those previous results with the present results in B cells, although it is noted that there was no great difference in the transcriptome of non-B cells in this study between the WT and ΔmiRNA-infected animals (which might therefore differ from the cow).

Reviewer #2: • Fig 2C: are there common genes in the two groups? Or is the profile of non-B and B cells completely different?

• Fig 2D: same comment as for 2C. Also, do the two viruses differentially downregulate host cell genes?

• Line 283-284: It would be interesting to show the effect of miRNA depletion on different phases of the virus life cycle. The manuscript would benefit from these data.

Reviewer #3: 1- the number of animals used for each experiment must be notified. Specially for fig 4-5-6 in which it is not clear how the stats were done.

2-p19 lane 407-408 the sentence is not complete and seems to be a repetition of the previous one.

3- I suggest to show the data obtained for #1131 animal in which infection with miRNA deletant resulted to high PVL. How is the B-cell proliferation in this animal ? Does the virus revert to wt ?

PLOS authors have the option to publish the peer review history of their article (what does this mean?). If published, this will include your full peer review and any attached files.

Reviewer #1: Yes: Charles Bangham

Reviewer #2: No

Reviewer #3: No
---

## [Editor Report · Decision Letter 1]

26 Mar 2020

Dear Pr Willems,

We are pleased to inform you that your manuscript 'Ablation of non-coding RNAs affects bovine leukemia virus B lymphocyte proliferation and abrogates oncogenesis' has been provisionally accepted for publication in PLOS Pathogens.

Best regards,

Bryan R. Cullen

Associate Editor

PLOS Pathogens

Susan Ross

Section Editor

PLOS Pathogens

Kasturi Haldar

Editor-in-Chief

PLOS Pathogens

orcid.org/0000-0001-5065-158X

Michael Malim

Editor-in-Chief

PLOS Pathogens

orcid.org/0000-0002-7699-2064
---

## [Editor Report · Acceptance letter]

1 May 2020

Dear Pr Willems,

We are delighted to inform you that your manuscript, "Ablation of non-coding RNAs affects bovine leukemia virus B lymphocyte proliferation and abrogates oncogenesis," has been formally accepted for publication in PLOS Pathogens.

Best regards,

Kasturi Haldar

Editor-in-Chief

PLOS Pathogens

orcid.org/0000-0001-5065-158X

Michael Malim

Editor-in-Chief

PLOS Pathogens

orcid.org/0000-0002-7699-2064